# “Virtual Surf Booth”: Assessment of a Novel Tool and Data Collection Process to Measure the Impact of a 6-Week Surf Programme on Mental Wellbeing

**DOI:** 10.3390/ijerph192416732

**Published:** 2022-12-13

**Authors:** Ariane Gerami, Charlie Foster, Joey Murphy

**Affiliations:** Centre for Exercise, Nutrition & Health Sciences, School for Policy Studies, University of Bristol, Bristol BS8 1TZ, UK

**Keywords:** acceptability, eudaimonic wellbeing, experience sampling method, feasibility, hedonic wellbeing, mental wellbeing, surf therapy, SWEMWBS, video, usability

## Abstract

Surf therapy is increasingly used as a health intervention, but evidence of its mental health benefits remains unclear. This longitudinal mixed-method study assessed the usability and acceptability of a novel online data collection tool and process to measure the impact of a surf programme on acute and chronic mental wellbeing. Fifteen women attending a 6-week surf programme in the UK were asked to complete a tool consisting of video recordings, word association and the Short Warwick-Edinburgh Mental Well-Being scale (SWEMWBS). Usability and acceptability were assessed through focus groups and quantitative data. The data generated in the focus groups, video recordings and word association were analysed via reflexive thematic analysis, and SWEMWBS presented descriptively. Participants perceived the tool as easy to use due to the completion time and its functionalities, and useful for self-reflection. Facilitating conditions such as timing and location, areas for improvement such as increased privacy, accessibility, incentivisation, and factors impacting data generated were further identified. Data collected covered both acute and chronic mental wellbeing and showed a positive relationship between surf and mental wellbeing. Further research is needed to confirm these findings in diverse populations, identify potential moderators, and confirm the validity of this tool and process.

## 1. Introduction

Being or practising a physical activity in blue space is increasingly shown as associated with an improvement in wellbeing [1,2,3]. Surfing is growing in popularity, with an estimated 35 million surfers in over 100 countries [4] and the discipline included in the 2020 Olympic games. Surf therapy is intended to improve health and wellbeing using surf as a vector [5]. It can be implemented as a stand-alone therapeutic intervention or incorporated into a wider programme to support health and wellbeing depending on the target population and practitioners involved, for example with mentoring sessions. Benninger and colleagues’ scoping review [6] reported the findings of 29 studies that had examined physical and psychological benefits of surf therapy interventions. The review reported significant improvements in health outcomes such as physical fitness levels, emotional regulation, social wellbeing, symptoms of anxiety and depression. However, they concluded that the level of evidence for the generalizability of the results remained low due to high variability in the groups and interventions. Another systematic review assessing health and wellbeing interventions delivered in natural blue spaces found that the available evidence supporting surf therapy health benefits was also highly heterogeneous [1]. Although the populations and interventions had high variability, the studies all reported improvements in the outcomes measured, whether mental or physical.

The mechanistic pathway through which surf therapy or any other physical activity could lead to an improvement in mental health remains unclear. A systematic review investigating the link between physical activity and cognitive and mental health in youth [7] did not find any strong evidence on the neurobiological or behavioural mechanisms. The strongest evidence was related to psychosocial mechanisms, where improvement in self-efficacy were obtained through improvement in physical activity self-perceptions, leading to enhanced cognitive and mental health. Being in blue and green spaces has also been shown to be linked to improved mental health, potentially mediated by being in a healthier environment conducive to social interactions, restoration, and physical activity [8,9,10]. The possible pathways of restoration, social connectedness and support, personal growth, physical activity, and feeling of being in a safe space have also been identified in the surf therapy research but their predictive weight remain to be confirmed [11,12,13]. More research is needed to elucidate the pathways of blue space and surfing leading to improved mental wellbeing.

There is no consensus yet about what constitutes mental wellbeing, how to best measure it and what theory should be the basis for policy making [14,15]. Although it may not capture all dimensions of mental wellbeing such as emotional wellbeing, life satisfaction and preference satisfaction [15], one prevalent theory considers mental wellbeing as having two dimensions, hedonic and eudaimonic [14,16]. Hedonism is defined as momentary pleasure [14], with hedonic wellbeing referring to feelings or moods with positive and negative affect [17]. Eudaimonia is seen to be where happiness and wellbeing can be reached by living in line with our own values to fulfil our true nature [18]. Whether hedonism is complementary to eudaimonia, or should one prevail over the other has been the subject of debate, however both dimensions are increasingly considered as complementary to define and reach mental wellbeing and happiness [14]. These two constructs were used in meta-analyses measuring the relation between nature connectedness and wellbeing [19,20], and were hypothesised to be linked to physical activity engagement [7], suggesting this theory may be relevant in the present study as well.

However, measurement of mental wellbeing can be challenging. Linton and colleagues [21] found 99 wellbeing measurement tools, with the majority being multidimensional, and 21 tools found to measure the mental aspect of wellbeing alone. Furthermore, for the measurement of hedonic wellbeing, one of the consistent challenges of studies is the recall and heuristic biases that are not fully mitigated by Experience Sampling Methods (ESM), such as the Ecological Momentary Assessment [22], and the Day Reconstruction Method [23], and are time-consuming for the respondents [22,24]. Ideally, studies could develop a method to assess the participants’ momentary mood while minimizing these biases when participants complete a survey sometime after the experience [25]. Alternative approaches are being developed to capture wellbeing data in the moment in a surfing context. Lisahunter and Stoodley used audio and video recordings to reconstitute the surfing experience [26]. Britton and Foley used “in situ” tools based on close interaction with the research team [27]. This novel tool could provide another alternative to the existing measurement tools used in the assessment of surf therapy programmes. As for any new measurement tool, there is a need to pilot this novel data collection method and process that intends to capture participants momentary mood part of the hedonic dimension, as well as the eudaimonic dimension of mental wellbeing, closer to the moment the activity (here surfing) occurs.

Thus, the primary objective of this study was to evaluate the usability and acceptability of an online data collection tool and process to evaluate mental wellbeing. A secondary objective was to assess the impact of the 6-week surf programme on participant acute and chronic mental wellbeing.

The study participants are women, from different ethnic backgrounds, some living in deprived areas. Gender, ethnicity, age, economic status, and cultural backgrounds have been identified as moderators in the use of and access to technology and perceived skills [28,29]. Therefore, it is hypothesized the feedback of this specific study population will help to improve the tool usability and acceptability for all groups of users. Furthermore, women being underrepresented and their presence in the surfing space not always valued [30,31,32], this study was a good opportunity to give them a voice.

## 2. Methods and Materials

### 2.1. Study Design

This study, conducted at The Wave, used an exploratory longitudinal cohort mixed-method design, whereby both qualitative and quantitative measurements were taken with the same study participants at defined timepoints to evaluate the impact of a surf programme [33]. The exploratory nature of this study is related to the primary objective of the study, that is to understand the participants perception of the data collection tool and process [34,35]. The Wave is an inland-surfing lake located in Bristol, UK, where waves are generated on demand every 8–10 s at different heights and speeds [36]. This study obtained ethics approval (no. 10951) from the School for Policy Studies Research Ethics Committee, University of Bristol, UK. Figure 1 summarizes the research questions and methods used in this study.

### 2.2. Study Population

This study used a purposive convenience sampling through a planned surf programme cohort at The Wave [36]. The programme participants were women from 3 community groups located in Bristol, UK, which focus on connecting diverse communities through outdoor activities, supporting refugees and asylum-seekers with community-building and wellbeing activities, and supporting individuals’ wellbeing through physical activity. The programme participants were approached by the lead author during their first session to introduce the study and invite them to take part. Their signed consent was obtained on this occasion. The process was facilitated by the management team at The Wave.

### 2.3. Programme Description

The programme consisted of weekly surf sessions for 6 weeks from May to July 2022. The first and second sessions were held two weeks apart due to public holidays. The participants were provided with all the equipment needed (wetsuit, helmets, colour-coded T-shirts indicating their experience level, aquatic shoes, and surfboard) upon their arrival.

As privacy was requested by the participants, only female surf instructors were involved with this cohort, and the sessions were held in a section of the lake where the participants mixed as little as possible with the public. The sessions were run by two surf instructors and lasted approximately 1.5 h, including 30 min of coaching out of the water for training and safety purposes. In the water, the participants were within their depth (able to stand up in the water) and could decide to take the waves as they come, helped by the instructors as needed. In line with the activities of the community groups involved in this cohort, the goal of the surf programme was to improve wellbeing through outdoor physical activity. There were no set individual or group goals, so the participants could set their own pace and rest on the shore whenever needed. The research team were present during each session.

### 2.4. Online Data Collection Tool

The tool provider was selected and the tool content was developed following the principles of the Technology Acceptance Model (TAM) [37], in particular the ease of use for the final end-users. The TAM is based on the Theory of Reasoned Action (TRA), where the dependent variable of behavioural intention (BI) is predicted by two main variables, perceived ease of use (PEOU) and perceived usefulness (PU). BI is a strong predictor of behaviour per the TRA. A meta-analysis of the TAM conducted on 88 studies in various domains demonstrated that the TAM is a reliable model to predict behavioural intention with the paths PU-BI and PEOU-PU having the strongest predictive coefficients (respectively, average β = 0.505 and 0.479, *p* = 0.000) [38].

The online tool was developed by Video Booth Systems, a company which provides video capture solutions for live events, and its development funded by the University of Bristol. The tool was hosted on a website accessible from a mobile device or computer and customized with a background picture of The Wave location. Images of the online tool and questions asked can be seen in Appendix A. The survey consisted of two video recordings, a word association question and a mental wellbeing scale. Before submitting, the participants had the opportunity to preview their videos and record them again if they wished. Video recordings were limited to two minutes. Upon completion of the last question, the entire survey could be submitted. Participants were able to skip questions (video recordings and word association) if they wished. The tool was piloted by the research team, the participants’ gatekeepers, and The Wave team before use, following the principles of questionnaire construction [39].

### 2.5. Evaluation of the Tool’s Usability and Acceptability

Use of the online tool was measured via the overall usage rate, the number of times the participants recorded the videos, the location (on-site, or at home) and the duration of the videos, and the number of words collected. Focus groups were held to understand and assess the participants’ perceptions of the data collection process and online tool. Considering the participants’ time constraints, the research team tried to minimize participants’ burden during the data collection phase. For this reason, the organization of virtual focus groups after the end of the programme was not deemed suitable. Upon the participants’ request and to maximize attendance, the focus groups were therefore held on-site before the week 5 session with all the participants on site. Drinks and food were offered to create an environment conducive to a discussion [40]. The semi-structured interview guide for the focus groups was developed using the TAM constructs, to discuss the participants’ PEOU and PU of the tool and its content, the data collection process (location, timing), the reasons for non-use, and their suggestions for improvement.

A focus group was held after the week 6 session with The Wave team to triangulate the study findings, and gather their perceptions about the extent to which the data collection interfered with the programme activities.

For all focus groups, discussions were elicited with the presentation of a graph showing the changes in mental wellbeing between week 1 and 4 and a draft of the word cloud generated from the word association question [40]. All interviews were conducted by the lead author.

The focus group interview guides are available in the Appendix A.

### 2.6. Assessment of Impact on Mental Wellbeing

The impact of the surf programme on mental wellbeing was measured with both qualitative and quantitative components, known as intramethod mixing [39]. It was hypothesized that the acute impact on mental wellbeing could be assessed qualitatively via the video recordings and words associated with the experience, while the chronic impact could be measured through a mental wellbeing scale. Although video recordings and word association are not validated to measure mental wellbeing in this context, one potential advantage, over researcher-led interviews, could be to reduce experimenter bias effect, whereby the researcher may subconsciously influence the participants’ answers [41], and the Hawthorne effect [42,43], whereby the study participants modify their behaviour as a result of their participation to a study. The video recordings were prompted by the following two open-ended questions: “What is your most memorable moment from The Wave this week?”, “Describe your feelings after visiting The Wave this week” and preceded by tips for recording. The word association was prompted by the following open-ended question: “What 3 words come to mind when thinking of your experience at The Wave?”.

The mental wellbeing scale selected for this study was the Short Warwick-Edinburgh Mental Well-Being Scale (SWEMWBS) [44], a self-administered 5-Likert scale composed of seven positively worded questions, asking participants to reflect on their last two weeks. The SWEMWBS is the short version of the Warwick-Edinburgh Mental Well-Being Scale (WEMWBS) [45]. WEMWBS was developed by the Universities of Warwick, Edinburgh and Leeds in conjunction with NHS Health Scotland. A score of less than 20 on the SWEMWBS scale categorizes a participant as having low wellbeing, between 20 and 27 as moderate wellbeing, and above 27 as high wellbeing. The maximum score is 35. The use of WEMBWS and SWEMBWS is recommended for research and evaluation purposes in surf therapy [46]. Furthermore, the tool has shown good content validity (Cronbach alpha = 0.91) [45], including amongst several English-speaking ethnic minority groups living in the UK (Cronbach alpha > 0.91) [47], good test–retest reliability at one week (0.83, *p* < 0.01), low susceptibility bias and no ceiling effect, making it an acceptable measurement tool to evaluate the impact of public health interventions at the population level [45]. However, it should be noted that the absence of a ceiling effect may depend on the context in which the SWEMWBS is used, as suggested in a cross-sectional public health survey conducted in Sweden [48]. It has validated translations available which was important for this study population. A validated translation in Arabic was provided to participants requiring it. The sensitivity to change of the SWEMBWS in the general population or in interventions involving physical activity remains to be confirmed [49]. The SWEMBWS covers mostly psychologic and eudaimonic wellbeing components, although a few items also cover hedonic wellbeing [44], which complements the type of data that were expected from the video recordings and word association questions.

The mental wellbeing scale was selected, and the video recording and word association questions were developed and their face validity (that is, the extent to which they appear they can measure mental wellbeing [50]) confirmed after reviewing the literature on surf therapy and mental wellbeing by the research team, consultation with a research expert in the field and with The Wave management team for their experience on previous programmes and research initiatives.

### 2.7. Data Collection

Demographic information, including age, gender, ethnicity, and nationality, were collected from the study participants at the start of the study. Given the mental wellbeing scale timeframe and to alleviate participant burden, the survey was administered to the participants after the session on weeks 1, 2, 4 and 6. Participants were given the opportunity to complete the survey on-site through laptops at the end of the session after they got changed, to collect data as close as possible to the session. The data collection was performed in the communal areas of the facility due to the need for a Wi-Fi connection. The lead author ensured there was enough space for the participants to keep their video recordings and answers to the survey private. The lead author explained how to use the tool and stayed in proximity to address any technical problems and answer questions. Finally, field notes were collected throughout the data collection period, including participant attendance, session duration, weather conditions, surf instructors, and on-site events during the session.

As the primary objective of the study was to assess the data collection process and online tool, the lead author decided to integrate participants into the study even if they joined the programme after the first week. Additionally, despite the impact on internal validity, the lead author also decided partway through the data collection period to allow data collection at different time points than planned in the study design when participants missed sessions, to increase the volume of data collected.

### 2.8. Data Analysis

The data analysis was conducted based on the framework proposed by Onwuegbuzie and Teddlie [35] for the analysis of mixed-method studies. This framework provides a procedure to choose analytic techniques based on considerations and decisions taken during the pre-analytic phase.

#### 2.8.1. Primary Objective

To address the primary objective, the goal was the triangulation and complementarity of data [35,51,52], where it was expected that the quantitative results inform the qualitative research question, and bring clarity to whether or not this novel online tool and data collection process was suitable to collect mental wellbeing data in this context and specific population.

Use of the online tool was analysed via quantitative content analysis [53] and descriptive statistics. Usability and acceptability were assessed through the focus group data. The discussions from the focus groups were audio recorded, transcribed and the transcription quality checked by the lead author. The data were analysed with elements of an inductive and deductive reflexive thematic analysis [54]. The coding was performed using a deductive approach [54] based on the TAM and the Unified Theory of Acceptance and Use of Technology (UTAUT) [55], and inductive approach to ensure that additional themes of interest outside of the framework were captured. The UTAUT is based on a review and integration of eight different models including the Theory of Reasoned Action, Theory of Planned Behaviour and the Social Cognitive Theory, that have proved to be relevant in a public health intervention context. Application of the UTAUT to public health interventions remains to be investigated, however it brings additional determinants that were worth being investigated in the context of this study such as social influence and facilitating conditions. In the context of this study, PEOU was defined as “the degree to which a person believes that using a particular system would be free of effort” [37], and PU as the “personal outcome expectations related to the performance of the behaviour” [55,56]. The development of themes through thematic analysis was done in an investigative way. The typology was then justified empirically, and using the TAM and UTAUT as referential [35]. The coding was performed by the lead author using Word and Excel, checked and agreed upon by another author (C.F.).

#### 2.8.2. Secondary Objective

Concerning the secondary objective, the goal was the expansion of data [35,51] where different types of media were used to collect different measurements of the surf programme impact on mental wellbeing. The hedonic and eudaimonic wellbeing data collected were not compared for triangulation as they were assumed to be complementary components of mental wellbeing. Therefore, the results were reported and interpreted separately. The videos recordings were transcribed, and the transcription quality checked by the lead author. The data collected through the video recordings and word association were coded and analysed with elements of an inductive and deductive thematic analysis [54]. The data identified as belonging to the mental wellbeing domain were coded according to the classification of hedonic and eudaimonic wellbeing definitions proposed by Huta and Waterman [57].

To generate the word cloud, the data collected through the word association question were grouped by similarity and etymology (e.g., ‘exciting’ and ‘excited’ were coded as ‘excited’). The coding was performed by the lead author using Word and Excel, checked and agreed upon by another author (C.F.).

The SWEMWBS data are presented descriptively (SPSS V28), and analysed quantitatively to assess the difference in the mean score of mental wellbeing between the first and the last measurement in the programme using the paired samples *t*-test and Wilcoxon signed-rank test Excel worksheet required by the SWEMWBS authors [58]. However, it should be noted that these tests could not be used for this study in theory since the distribution of differences between the two paired groups was not normal or symmetrical [59,60]. For this reason, the results of the paired-samples *t*-test and Wilcoxon signed-rank test were only included in the Appendix A.

## 3. Results

### 3.1. Study Population

15 out of 19 programme participants (79%) took part in the study. One declined participation due to her perceived low self-efficacy with technology. The reasons for the decline of the other programme participants are unknown.

The study participants are all women (N = 15), with a mean age of 43 years old (SD 5), and the majority being from the white ethnic group (N = 6, 40%) and British nationality (N = 8, 53%). The study participants’ demographics are presented in Table 1.

### 3.2. Programme Attendance

The study participants attended an average of 4.1 sessions (SD 1.6) out of the 6 planned. Ten of them attended at least four sessions. Two participants dropped out of the programme due to privacy concerns, with other dropout reasons unknown.

### 3.3. Primary Objective—Usability and Acceptability of the Tool

Thirteen participants completed the survey at least once, and two never completed the survey. In total 35 surveys were collected. The survey was completed on average 2.3 times (SD 1.5) out of the four timepoints initially planned. Out of the ten participants who attended at least four sessions, the survey was completed 3 times on average (SD 1.1). Out of the 11 occurrences where the online tool link was emailed to participants upon their request to complete the survey at home, the survey was completed 3 times (27%). The descriptive statistics of the tool use are presented in Table 2.

#### 3.3.1. Perceptions and Use of the Tool and Data Collection Process

Two focus groups were held with 11 study participants. The focus group with The Wave team included three participants that were in regular contact with the programme participants. The surf instructors could not participate, or their feedback being collected afterwards due to time constraints. The focus groups lasted between 20 and 30 min each.

Several themes about the usability and acceptability of this tool and data collection process emerged from the focus groups and the quantitative analysis of the tool use. Several themes were related to the TAM and UTAUT constructs.

##### The Tool Was Perceived as Easy to Use

The tool was perceived as easy to use, mainly because it was perceived as quick to complete: “it’s quite quick… In that sense it’s quite easy” (Participant no. 7), “I like the fact that it was short, it wasn’t too long, it was just right. You could do it” (P13). The PEOU increased with their experience of using the tool: “I think I got used to it over the first weeks. After a few times you’ve done it, it’s not felt so awkward. I feel like I knew what to expect” (P7). The balance of duration versus frequency was perceived as acceptable, although they would find it easier if this was done weekly.

The PEOU was also supported by the existing functionalities of the tool, “Because it does most of it for you, you really don’t have to do much. It’s quite simple” (P1). This was especially true for the video recordings when participants were not comfortable with this media: “I like the fact that you could delete and do it again” (P14).

One participant found the video recording easier to use than typing: “The video is good for that, cause if you tried to type it’s not as spontaneous” (P9).

Lastly, the questions were perceived as “easy” to answer, because “the language was pitched so it was accessible” (P9), and because “those questions are being normalised more. People are becoming more and more used to seeing those questions” (P11). To further increase the PEOU, it was suggested emphasizing the survey duration and the mental wellbeing scale timeframe in the tool.

##### The Tool Was Perceived as Useful

The survey was perceived as useful because it provided an opportunity for self-reflection that is not always available in their daily routine:

“It’s quite nice to get the opportunity to actually tell how I am really feeling”.(P7)

The participants appreciated and were engaged when presented with the draft outputs from the data collected, which supported the PU of the tool and survey. To increase the PU and survey completion in future use, participants suggested that the idea of self-reflection should be used in the tool signage, feedback could be provided to the participants via a live word cloud, and the survey should capture more in-depth data over time. Being able to share a picture on social media through the tool was mentioned as an incentive, with an emphasis on it being free, fun and social.

##### Facilitating Conditions to Increase to Tool Use Were Identified

The facilitating conditions evoked by the participants would support an increase in the intent to use the tool and its actual use, and impact the amount of data generated. Survey completion on site should be encouraged and supported, “otherwise as soon as people leave that’s it, their brains, they’re back into everything else” (P9). The researcher’s proximity and connectedness with the programme participants were identified as a facilitating condition by the participants and by The Wave team, suggesting that having a person on-site promoting the survey and the use of the tool is important. Additionally, having the equipment on site to complete the survey was also perceived as a facilitating condition.

“actually having the [laptops] here and everybody feeling as though they can contribute [was] really helpful”.(P11)

Although most of the participants preferred to complete the survey on site straight after the session as shown by the quantitative data, the choice to complete it at home is still warranted to facilitate different types of participants.

“when I did it at home, I really, I felt my answers […] more comfortable for me”.(P13)

The level of perceived privacy was also evoked as a factor that would impact their answers.

“you get no distraction or anything so you could be more open”, “If I had been in a booth, I would probably have taken more time thinking about what to say”.(P7)

The idea of having a dedicated space on-site to complete the survey was welcomed. This space would need to provide privacy within a low-noise and low-traffic environment, especially to reduce the possibility of being overheard and made feel uncomfortable when recording videos.

“Big thing for me would be to have a space with no one behind you. Not necessarily that people can hear you, but so there is no one around you”.(P7)

Although the tool was perceived as easy to use, there were several suggestions to make it more accessible. To increase PEOU, the participants should be given the choice of media to use (e.g., audio recording or paper survey as an alternative). Participants uncomfortable with expressing themselves, typing or spelling could be provided with word suggestions.

“the word map, I wonder if you could turn that into like a live thing, so people could click on them? Like you say, if it’s prohibitive finding or spelling or whatever, you could choose”.(P9)

##### Social Influence May Impact the Use of the Tool

Social influence was also identified as a determinant in the intent to use the tool, with a higher study participation acceptance during the presentation to the initial group, than when individuals joined the programme after the first week. Furthermore, observations throughout data collection identified that seeing participants completing the survey on-site seemed to remind and encourage others to do so as well. Providing privacy was seen as a facilitating condition that would also decrease the impact of social influence on the participants’ answers.

Beyond this, the social norms were not perceived as having an impact on the data generated. The participants, when comfortable with the video media, were not self-conscious with their physical appearance on the video recordings: “there is no expectation you know, not just for Friday night…” (P9), “I actually quite like looking at it, cause I look like I am quite really overexcited, […] like I had the nicest time” (P9).

##### Self-Efficacy May Impact the Perceived Ease of Use

Low perceived self-efficacy with digital tools was also identified as a factor in the PEOU:

“for those who are not computer savvy” (P14), “Most of us are not [very technical]”.

Implementing facilitating conditions would compensate for lower self-efficacy with digital tools, and therefore increase PEOU.

#### 3.3.2. Factors Impacting the Data Generated

Several additional factors influenced the data collected through the online tool. Although participants were given the option to complete the survey after leaving the site, this was not always possible due to their other commitments. One participant mentioned that “[they] were a bit like hurried up, because [they] had to all go” (P7) and that she “definitely had to give short answers as [she] wanted to quickly get through it” (P7), suggesting that the lack of time had an impact on the depth and breadth of answers provided. Although self-reflection was perceived as an opportunity, the nature of the survey and the questions could also trigger difficult feelings.

“Yes, so it’s just, actually, sometimes those questions, although you could think, ‘Oh, they might trigger some kinds of people’, really, actually, they might help also, that people to understand, actually, this is a positive for me”.(P11)

Events during the sessions, even minor, could also impact the answers to the survey. Two participants having experienced minor injuries skipped the video recording on this occasion, although they completed the rest of the survey without a noticeable impact on their answers.

Lastly, participants were made aware of the study purpose, and expressed that they were already convinced of the positive impact of this surf programme on mental wellbeing. One participant recognized the existence of a research-effect bias and tried to mitigate this at her level.

“I think sometimes I was a bit… not biased, but I was like I really want this research to work”.(P6)

For these reasons, the answers provided to this survey may not be a full and accurate reflection of the participants’ mental wellbeing state.

### 3.4. Secondary Objective—Impact on Mental Wellbeing

#### 3.4.1. Video Recordings

As expected, the videos captured the participants’ acute or ‘in the moment’ feelings. The participants mentioned changes that they perceived in their mental wellbeing, the physical impact of the surf session, or the events causing the feelings expressed.

##### Acute Feelings of Wellbeing

The feelings expressed are characteristic of positive affect in the hedonic wellbeing dimension, with quotes such as “I feel really happy”, “exhilarated”, “positive”, “energized”, “proud of myself”, “my mood has lifted”, “having that amazing buzz when you catch a wave”, “being in the water, it’s just sort of emptied the mind of anxiety”, “walking up the walk from the car park to the wave I started to get sort a lot of butterflies”, “[…] that buzz about doing something completely different out of the routine”, “watching all the women smile and have a laugh” mentioned. Where negative affect was expressed, it linked to physical sensations (e.g., “really tired”), the end of the programme (e.g., “I guess I felt a bit sad this week because it was my last week”), or events during the session (“I did not really focus on being present and being very mindful”).

Participants also expressed feelings characteristic of eudaimonic wellbeing and linked to the experience. Feelings of individual or collective growth and sense of achievement, effort and social connectedness were identified, with participants “learn[ing] something new each time”, sharing “the unified goal of standing up on these boards”, experiencing an “amazing feeling to finally being able to feel confident to stand up”, and “seeing everyone’s faces just with the improvement and the confidence”.

##### Changes in Mental Wellbeing

Participants expressed that they noticed a change in their mental wellbeing over time, either in the hedonic or the eudaimonic wellbeing domains. Participants expressed that they felt “a bit more mindful, more so than last week”, “much calmer”, “much better this week”, and reflected on “how heavy [their] thoughts have been in the last few weeks, and now [being] just absolutely buzzing”.

##### Physical Impact

Another emergent theme in these video recordings was the physical impact of the surf session. The participants described a positive impact, with participants feeling “a bit achy at the moment but really good”, “happy tired”, “exhausted in like a brilliant way that only this sort of exercise can give you”. Participants also felt like “[they]’ve worked out”, “[had] a lot of energy” and were “managing to do 23,000 steps that day”.

##### Weather Impact

The weather was mostly favourable during the sessions. The sunny weather was evoked as a reason for positive affect. The less favourable weather was evoked in the participants’ in a positive way: “even though it was raining, and the weather wasn’t that great it did not put anyone off”. The weather during the session did not seem to impact the participant’s acute feelings to a significant extent. Weather conditions are reported in the Appendix A.

#### 3.4.2. Word Association

Out of the 100 words collected, 81% are evocative of hedonic wellbeing feelings such as happiness, excitement, satisfaction, relaxation, etc. 13% of the words are related to eudaimonic wellbeing, covering the constructs of growth, mindfulness, effort, and self-realization. The last 6% evoke the physical impact of the experience. The results are presented in Figure 2. The frequency of the words is reflected by their size on the word cloud.

#### 3.4.3. Changes in Mental Wellbeing Score (SWEMWBS)

The descriptive statistics are shown in Table 3.

The SWEMWBS mean score was 24.3 (SD 7.0) at the first measurement, and 28.5 (SD 4.2) at the last measurement.

## 4. Discussion

The purpose of this study was to assess the usability and acceptability of a novel online tool and data collection process to evaluate the impact of a 6-week surf programme on acute and chronic mental wellbeing. The tool was used on average 2.3 times out of the average of 4.1 sessions attended by the study participants and of the 4 measurement timepoints planned, a completion rate of 57.5%. This is lower than observed in ESM using mobile devices (average of 69.6%, SD 22.8) [61]. This is partly explained by the fact that participants may have missed the surfing session where data collection was planned. The participants may have found attending all sessions of the programme challenging, but when they attended they had no issue in completing the tool. The use of the tool could be increased by administering the survey weekly at each session to decrease the missing data linked to missed sessions, and will be consistent with existing surf therapy research [62,63]. It is important to encourage participants to complete the survey while still on-site to capture their momentary mood. However, it is also important to offer the possibility to complete it at home as pointed out in the focus groups, acknowledging this would impact the validity of the data collected to assess the acute mental wellbeing.

The themes emerging from the focus groups show that the tool was perceived as quick and easy to use despite a relatively low perceived self-efficacy with digital tools, and useful as a means for self-reflection. The use of this tool could be further increased by enhancing its PEOU and PU and implementing further facilitating conditions, especially increasing privacy when completing the video recordings on site. It should be highlighted that providing word suggestions to increase accessibility, as suggested during the focus groups, would come with potential bias from a research point of view, as the participants may choose words that look more socially desirable but do not truly represent their views. Several factors impacting the data generated by the participants, such as lack of time, events during the sessions, potential research bias, or mental wellbeing state were further identified.

The data generated through the video recordings, word association and SWEMWBS show that these are potential complementary measurement tools to assess the different dimensions and constructs of mental wellbeing. There was moderate redundancy between the themes emerging from the video recordings (mainly positive affect related to the experience related to hedonic wellbeing), and constructs measured via the SWEMWBS (mainly measuring hedonic wellbeing through feelings of optimism, relaxation and evaluative mindset, and eudaimonic wellbeing through relationships, autonomy, competence) [45]. There was a higher redundancy of themes between the video recordings and the word association. However, the word cloud is a valuable output as it can provide feedback to the participants that could increase the perceived usefulness of the online tool. This complementarity in the mental wellbeing construct measurements can strengthen the validity of the results of the impact of surf therapy on mental wellbeing in further research [39]. Overall, the online process enabled the collection of data through various media and provided data on acute and chronic status for several constructs of mental wellbeing. This method could be used to assess the impact of several physical activities on mental wellbeing, such as brisk walking, running, swimming [64], where both the acute and chronic effects are important to be investigated and could be affected by recall bias. However, it should be recognized that there may be limitations for its dissemination to other settings and countries due to the need of a device and Internet connection in its actual form and its relative higher cost compared to paper-administered tools. Lastly, although this study aimed primarily to evaluate a novel tool, some of the findings presented here could be transferred and used in further surf therapy research independently of the tool.

Future research should seek to investigate the usability and acceptability in varied study populations and explore further the reasons of non-use to ensure that this tool and data collection process are adapted to measure mental wellbeing outcomes in the context of this surf programme. Additionally, there is a need to assess the validity of the measurement tools to evaluate the impact of surf on mental wellbeing in a population more representative of the targeted population, before assessing their experimental validity, as suggested in the proposed draft Edinburgh Framework for quantitative measurement research [50].

Regarding the impact of the surf programme, the data generated through the video recordings support the causal relationship between the surf session and the acute feelings of mental wellbeing. This included feelings of positive affect, social connectedness, growth, and sense of achievement. The quantitative results also show an improvement in mental wellbeing with an increase of the SWEMWBS score from a mean of 24.3 (SD 7.0) to 28.5 (SD 4.2). These results are similar to those found in the existing surf therapy literature. Marshall and colleagues [11] found increased feelings of accomplishment, sense of respite, sense of mastery and social connectedness from surfing experiences in 22 young people. Furthermore, a mixed-method evaluation of a 6-week surf programme “The Wave Project” with 121 young people at risk of mental health issues or social exclusion, increased feelings of positive effect and sense of achievement [65,66]. Participants also showed improvements in positive functioning (t(82) = −6.42, *p* < 0.000) and emotional wellbeing (t(81) = −3.87, *p* < 0.000) measured by the validated Stirling Children’s Wellbeing Scale [65,66]. These findings are also supported by the sense of achievement and the positive impact on mood induced by surf sessions expressed in a mixed-method study conducted with 9 young people [13]. The study also showed an increase in the means of mental health and social connectedness outcomes using validated tools, although no statistical analysis was run due to lack of power.

However, a positive relationship and causal inference between the surf programme and the improvement in mental wellbeing cannot be ascertained from the quantitative results in the present study due to non-random sampling, no pre-intervention measurement, no measurement of confounders such as physical activity and socio-economic levels affecting the internal and external validity [33]. Further research is needed to provide evidence of this relationship. Additionally, although the SWEMWBS showed no ceiling effect at the population level [44], the participants’ scores were distributed with a negative skewness, suggesting a possible ceiling effect [67] of the tool in this study. The measurement tool in future studies should be selected depending on the intervention or programme evaluation and on the target population to ensure it is based on the hypothesized pathways between independent and dependent variables, and is sensitive enough to change in order to assess an impact [48,50]. Another area for further investigation will be to investigate the sustainability of the change in chronic mental wellbeing after the end of the programme.

The constructs of the TAM and the UTAUT emerged from the qualitative data. Further research is needed to establish the predictive values of these variables in this context. There is a lack of literature about intramethod mixing used in health intervention evaluations which prevents an in-depth discussion in relation to the existing knowledge base. However, for all the moderators studied in the TAM and in the context of ESM using mobile devices [61], PU was the strongest predictor of BI. Therefore, further efforts should focus on measuring and increasing the PU of this tool in various populations.

A strength of the current study resides in the fact that it was conducted in free-living conditions, which may provide better information and informed decisions on whether the tool has good reliability [50]. The concurrent triangulation design strengthened the findings of the study, where qualitative and quantitative results were combined through a parallel mixed analysis to extract more meaning from the data than would be obtained by analysing the two datasets separately [52,68,69]. In future research, the qualitative and quantitative measures of mental wellbeing could also be measured and analysed differently (e.g., acute mental wellbeing measured quantitatively and chronic mental wellbeing measured qualitatively) to explore further this area.

It is unavoidable that the interpretation of the data is influenced by the researcher’s background and beliefs as for any qualitative analysis [34]. To aid credibility of this research and ensure a true representation of the data, the lead author used reflexivity to recognize how their values and views may influence the findings [70], by writing field notes during and after each session documenting the events, the participants’ behaviour and comments, the surf instructors’ comments, and the lead author’s feelings in relation to these.

However, several limitations affect the credibility and transferability of the results.

The first focus group may have been too large and diverse to allow all participants to express their views, thereby limiting the breadth and depth of information collected [40]. In addition, some non-users of the tool either did not participate or speak during the focus group meaning their perceptions were unknown. For these reasons, additional views as to why the online tool was not acceptable may have been missed.

Although the research team implemented different measures to increase credibility (prolonged engagement with the participants and observations on-site, data and method triangulation), one main limitation is the lack of investigator triangulation and member checking due to time constraints, whereby the coding was performed by one researcher and the transcripts and coding were not checked by the participants [71,72].

Furthermore, the credibility of the results related to the primary and secondary objectives may have been affected by the participants’ existing involvement in physical activity programs to improve mental wellbeing, and by the interviewer effect, where the participants relate to the interviewer’s background [39,73]. Both phenomena may have impacted positively their perceptions of the tool and the data collection process, or increased the PU and therefore the use of the tool, which may not be replicated in another study.

There may have been an element of social desirability bias in the answers to the survey, especially in the video recordings where the participants were aware that the research team will watch them. Lastly, due to a purposive convenience sample, there may be little transferability of the findings outside of this study population. The PEOU, PU, BI, actual use and data generated are likely dependent on age and gender [55].

Another limitation of this study is that the participants’ level of self-esteem was not measured. As some participants found their self-reflection in the video recordings uncomfortable, it is possible that self-esteem could have an impact on the breadth and depth of data generated via the video media.

Furthermore, although the participants freely agreed to participate in the study, it is possible that not all participants have the willingness (be prepared to) or eagerness (strong internal drive) to share their feelings [74].

## 5. Conclusions

This study provides evidence that this intramethod tool and data collection process were perceived as easy to use and useful by a group of women and could be further improved for future use. This tool and data collection process allowed the acute and chronic mental wellbeing to be evaluated closer to the time of the experience, and captured data related to both the hedonic and eudaimonic dimensions of mental wellbeing. The data indicates that mental wellbeing may be supported by participating in the surf programme. However, further research is needed to confirm these findings in diverse populations, identify potential moderators impacting the use of the tool and the data generated, and confirm the validity of this tool and process for experimental use.

## Figures and Tables

**Figure 1 ijerph-19-16732-f001:**
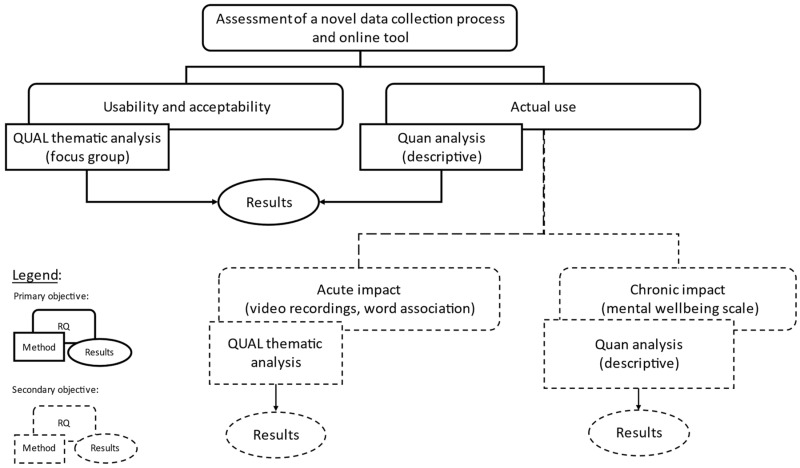
Research questions (RQ) and methods.

**Figure 2 ijerph-19-16732-f002:**
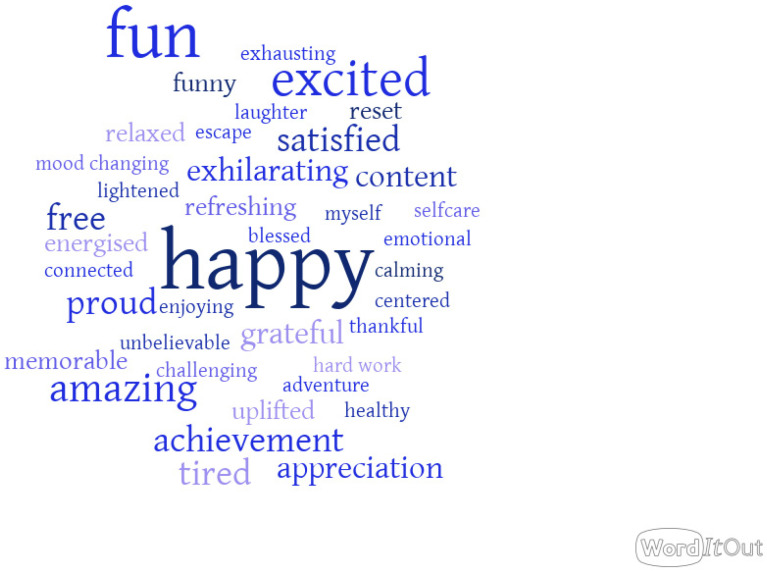
Word cloud for “What 3 words come to mind when thinking of your experience at The Wave?”.

**Table 1 ijerph-19-16732-t001:** Study participants’ demographics.

**Variable**	**Mean (SD)**
Age	43.0 (5.0)
**Variable**	**N (%)**
Gender	
Female	15 (100)
Male	0 (0)
Ethnicity	
Asian/Asian British	2 (13)
Black/African/Caribbean/Black British	5 (33)
Mixed/multiple ethnic groups	1 (7)
Other ethnic group	1 (7)
White	6 (40)
Nationality	
Bangladeshi	1 (7)
British	8 (53)
British Pakistani	1 (7)
English	1 (7)
Nigerian	2 (13)
Somalian	1 (7)
Syrian	1 (7)

Abbreviation: SD: Standard Deviation.

**Table 2 ijerph-19-16732-t002:** Tool use—descriptive statistics.

Reference Population: Study Participants (N = 15)	Total	Mean	SD	Median	Min	Max	%	% Reference
Programme attendance								
Number of sessions attended	62	4.1	1.6	5	1	6	69	Sessions planned per programme (N = 90)
Survey completion								
Number of surveys completed *	35	2.3	1.4	2	0	4	100	
Survey location/timing								
On-site/straight after the session	31	-	-	-	-	-	89	Surveys completed (N = 35)
At home/day of the session	1	-	-	-	-	-	3	Surveys completed (N = 35)
At home/another day	3	-	-	-	-	-	9	Surveys completed (N = 35)
Video recordings								
Completed	29	-	-	-	-	-	83	Surveys completed (N = 35)
Audio only **	1	-	-	-	-	-	3	
Duration (Q1 & Q2) (in seconds)	-	19	14	16	3	79	-	
Skipped **	6	-	-	-	-	-	17	
Word association								
Completed	34	-	-	-	-	-	97	Surveys completed (N = 35)
Number of words ***	100	2.9	0.6	3	0	3	-	
Number of unique words	50	-	-	-	-	-	-	
Skipped **	1						3	Surveys completed (N = 35)
Mental wellbeing scale								
Completed ****	35	-	-	-	-	-	100	Surveys completed (N = 35)

* Survey completion planned at 4 timepoints (Week 1, 2, 4 & 6); ** In these instances, the survey was completed on-site; *** 3 entries maximum; **** Completion mandatory to submit the entire survey.

**Table 3 ijerph-19-16732-t003:** SWEMWBS scores—descriptive statistics.

		First Measurement	Last Measurement
N		11	12
Mean		24.3	28.5
Median		27.0	29.0
Standard Deviation		7.0	4.2
Range		21	15
Minimum		10	19
Maximum		31	34
Percentiles	25	19.0	27.0
	50	27.0	29.0
	75	29.0	31.0

## Data Availability

Available upon request to the lead author.

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
