# Peer review of "“Virtual Surf Booth”: Assessment of a Novel Tool and Data Collection Process to Measure the Impact of a 6-Week Surf Programme on Mental Wellbeing"

_ijerph, 2022, doi:10.3390/ijerph192416732_

Round 1

Reviewer 1 Report

The manuscript describes a novel, online data collection tool for brief qualitative and quantitative measures that evaluate the impact of surf therapy sessions on the participant’s affect and experiences. Results indicated that among those who used the tool, they generally found that it was easy to use. The manuscript is well-written and presents a new tool for potential use, which could contribute to the literature. Additionally, the supplementary documents (and particularly the tool screenshots) were helpful. However, although the authors’ state limitations of the study, content may need to be tempered in sections and the limitations acknowledged throughout the manuscript. Suggestions to strengthen the contribution of the manuscript are described below:

Main concern:

The primary concern with the manuscript is the framing of the Introduction. The way that it is written, it seems as though this tool is going to revolutionize data collection and address gaps with current approaches. The authors particularly focus on how this tool can improve in-the-moment data collection. However, when you take into account the benefits, cost, and actual data – this measure seems to be subject to common problems inherent in other approaches and in some ways, is less compelling than existing measures. For example, there are several surf therapy studies that use the well-established PANAS (or PAS and NAS separately) on the beach before and after weekly surf therapy using paper and pen and have higher rates of completion (recognizing this addresses the quantitative, but not qualitative data collection). The online tool is not only more expensive, but it requires some technical knowledge, a device, and Wi-Fi, which is often not available on beaches in industrialized countries, let alone developing countries. This is an important consideration given that surf therapy is conducted worldwide. The authors should exercise caution in their promise for this measure because although it is novel and could be useful, usage rates were fairly low, there are significant limitations for dissemination, and it does not appear to build upon lower cost/resource (quantitative) options currently available in the published literature. A more accurate framing of the tool is that it is novel and may be suitable for certain populations and settings.

Another main concern is that the authors use significance testing for the pre- to post-findings. With only 7 participants, it seems that this is feasibility/pilot data and therefore, significance testing is inappropriate. The authors should consider reporting the means/SDs descriptively but removing content about significance testing.

Minor concerns:

Please briefly define surf therapy in the first paragraph of the Introduction to set the construct apart from mentions of surfing earlier in the paragraph.  

Although there are limited data on mechanisms for why surf therapy/physical activity interventions result in improved psychological and physical benefits (especially quantitative), it might be helpful for the authors to at least theorize potential mechanisms (e.g., social interaction, being outdoors, purposeful activity, etc.) or cite qualitative findings that could be explored quantitatively in future research (lines 42-49).

The authors emphasize that the tool can be used specifically for in-the-moment data collection, but how does completing the assessment at home meet this goal?

The authors’ state that the acute findings could be explored qualitatively and that chronic outcomes could be evaluated quantitatively (section 2.6); however, it is unclear why either outcome couldn’t be explored through either means. If there is a rationale for this statement, it would be useful to state it because arguably both approaches can be used to assess either outcome depending on the research question.

To protect privacy and confidentiality of study participants, the authors may want to collapse categories where there is/are only 1 or 2 participants.

This is just a comment – participants mentioning that conducting the assessments weekly might facilitate greater completion is interesting given that it is consistent with existing surf therapy research evaluating the effects of weekly surf sessions on symptoms and affect with high completion rates.

It was notable that many of the comments provided by participants are about conducting assessments on the beach more generally and not specific to this tool. The authors should emphasize this aspect as it would be useful information for readers wanting to implement assessment in surf therapy programs and to advance the methodology and science of surf therapy.

Another comment – It was observed that the baseline scores of participants were quite high on the SWEMWBS. Is this a higher functioning sample? Or does this raise questions about the utility of the measure and potential psychometric limitations regarding sensitivity to change? It was also noted that although there was statistically significant change from pre- to post-session on this measure (please see comment above about not conducting significance testing, however), it appears that only 3 had meaningful change. Despite mentioning that prior research has suggested that there may not be a ceiling effect for the SWEMWBS, the authors appropriately mention that their results could present data to the contrary.

Reviewer 2 Report

Thank you to the authors for a very interesting paper. A very interesting concept and read. I have a few minor considerations listed below for you to consider:

Section 2.8 Data analysis is just a statement. What about the framework proposed by Onwuegbuzie and Teddlie [28]? No context provided.

Typo in Table 1, Mead (SD) is this meant to be Mean?

Formatting of the paragraph in lines 547-548.

Typo line 552 "the online tool a was not acceptable"

Other than these formatting errors, the paper in general, and the concept itself was presented clearly.

Reviewer 4 Report

Thank you for the opportunity to review this manuscript. It is a fascinating read, looking at a novel method to examine cognition of females undergoing surf therapy. Below are some comments to potentially improve it

Abstract line 10: Is there a clearer word than heterogenous here? Perhaps “unclear” or “ambivalent” or “divided”? Whatever is most accurate

Intro: Line 35: Can you perhaps provide an example of a health outcome improved via surf here?

Methods: Line 93: do you mean from May to July?

Line 163: This sentence on content validity seems a repetition off line 160. Could delete one or the other

I did not see Figure 1 referenced in the text (perhaps I missed it?) and the figure could potentially be deleted. Not sure how much it adds to the manuscript

Line 193: Do you mean the framework was used, or guided this study? Missing a verb in the sentence. Could you briefly describe the framework?

Lines 217 and 230: Was the coding verified and agreed upon by the rest of the authors? If not, this could be a limitation

Table 1: Given the sample size is so small, no need for decimal places in the percent column. I suggest rounding to the nearest percent

Lines 350-352: This sentence sounds like it belongs in the discussion section

Lines 391-392: Had to comment on this…Very interesting! Never heard a research participant state that before

Table 3: Given the very small sample, perhaps include the median and IQR here since you provide the median difference in the text

Line 494: Could you provide an example physical activity where these tools might also work?

Line 545: Can you provide an example of how reflexivity was used?

Conclusions: Can you provide any conclusions specific to surf therapy and cognition? It might help tie everything together

Round 2

Reviewer 3 Report

The authors have made a thorough revision of the article and have met up on most of the point the was addressed as crucial for quality.

In the last peer review report i recommended rejection and presently the article is balancing for yes/no.

Why? - still the issue of Thematic analysis, and lack of analythical deepness...

Having read up on the references that the author uses to support their analysis. The perception is that the analysis lacks "analysis" and mostly represents a deductive sorting data to overarching domain. And yes, deducation is allowed in the method, but the method "developers "Brown and Clarke" has protested against to shallow use of the method in the way (that i perceive that you have used it) -- since it leaves the reader to make most of the analysis by reading "between the lines". As reviewer i can accept that all articles are not 100% quality, and since your article manages to make an argument for the objective relating to "usability and acceptablility".

With the help of all reviewers the article has made significant impovements, and i can recommend publication.

But i want to state that the addition in the conclusion does not have support in data as it is presently written (line 872-3). The quantitative data analysis shows impovements in scores - but the design does not support the idea of a causal link (it still may have!).

Suggestion: "The data indicates that mental wellbeing may be supported by participating in the surf programme"
